# Constant-Power versus Constant-Voltage Actuation in Frequency Sweeps for Acoustofluidic Applications

**DOI:** 10.3390/mi13111886

**Published:** 2022-11-01

**Authors:** Fabian Lickert, Henrik Bruus, Massimiliano Rossi

**Affiliations:** Department of Physics, Technical University of Denmark, DTU Physics Building 309, DK-2800 Kongens Lyngby, Denmark

**Keywords:** acoustofluidics, microparticle acoustophoresis, general defocusing particle tracking, particle-velocity spectroscopy

## Abstract

Supplying a piezoelectric transducer with constant voltage or constant power during a frequency sweep can lead to different results in the determination of the acoustofluidic resonance frequencies, which are observed when studying the acoustophoretic displacements and velocities of particles suspended in a liquid-filled microchannel. In this work, three cases are considered: (1) Constant input voltage into the power amplifier, (2) constant voltage across the piezoelectric transducer, and (3) constant average power dissipation in the transducer. For each case, the measured and the simulated responses are compared, and good agreement is obtained. It is shown that Case 1, the simplest and most frequently used approach, is largely affected by the impedance of the used amplifier and wiring, so it is therefore not suitable for a reproducible characterization of the intrinsic properties of the acoustofluidic device. Case 2 strongly favors resonances at frequencies yielding the lowest impedance of the piezoelectric transducer, so small details in the acoustic response at frequencies far from the transducer resonance can easily be missed. Case 3 provides the most reliable approach, revealing both the resonant frequency, where the power-efficiency is the highest, as well as other secondary resonances across the spectrum.

## 1. Introduction

In many experimental acoustofluidic platforms, the device is actuated by an attached piezoelectric transducer, driven by a sine-wave generator through a power amplifier. To describe the performance of the acoustofluidic actuation, the operating conditions are typically expressed in terms of the voltage amplitude or the electric power dissipation together with quantities such as the acoustic energy density, the acoustic focusing time, or achievable flow rates [1,2,3]. Often, it is however left unclear under which conditions and at which point in the electric circuit, the relevant quantities such as voltage amplitude or power dissipation have been measured. Recent studies compare device performance at constant average power for different placements of the transducer [4,5]. Dubay et al. [6] performed thorough power and voltage measurements for the evaluation of their acoustofluidic device, however, they noted that the actual power delivered to the transducer might reduce to only a fraction (as low as 10%) of the reported value. The likely cause of this reduction is that the transducer is acting as a large capacitive load, where electrical impedance matching between source and load impedance is difficult to accomplish [6,7].

Whereas optimization of the driving circuit is customary in other fields, such as ultrasonic transducers for cellular applications [8], non-destructive testing [9], and pulse-echo systems [10], this has not been given much consideration in the field of acoustofluidics, where the focus often lies on optimizing the acoustic impedance matching [11,12], while neglecting the impact of the driving circuit. A recent work, though, considers topics such as electrical impedance matching in the context of developing low-cost and possibly hand-held driving circuits for acoustofluidics [13]. To our knowledge, studies have not yet been performed, in which the impact of different electrical excitation methods on a transducer in a given acoustofluidic device is compared with respect to the resulting acoustophoretic particle focusing.

In the case of bulk piezoelectric transducers, where the electrical impedance ranges over several orders of magnitude as a function of frequency, the voltage amplitude across the transducer can differ severely from the amplitude expected by simply considering the voltage input at the amplifier. Suitable voltage compensation circuits or voltage correction methods should be used to achieve the desired voltage amplitude directly at the transducer. Furthermore, a standard has not yet been established whether it is more beneficial to run frequency sweeps at a constant voltage or at a constant power. We therefore in this work investigate the impact of three different actuation approaches during a frequency sweep: (1) Constant input voltage into the amplifier, (2) constant voltage at the transducer, and (3) constant power dissipation in the transducer. We compare experimental findings with our numerical model. The aim of this paper is to establish guidelines on which actuation approach is preferable for acoustofluidic applications using bulk piezoelectric transducers to generate acoustophoresis in bulk acoustic waves.

The paper is structured in the following way: In Section 2 a brief summary is given of the governing equations for the pressure field, the displacement field, and the electric potential in our acoustofluidic device. Section 3 gives an overview of our experimental setup, and the procedure used for the measurement of the particle velocities is described step by step. In Section 4 we describe the numerical approach used in our study, and in Section 5 we compare several aspects of the obtained results for the device under study: a comparison between the electrical characteristics of the device, as well as the numerically and experimentally observed acoustophoretic particle velocities are given. Furthermore, some details of the simulated fields are shown. Finally, the paper concludes in Section 6 with a short summary and some guidelines on the actuation of piezoelectric transducers for acoustofluidic applications.

## 2. Theory

The theoretical approach follows our previous work [3,14,15,16], in which the computational effort in the simulations is reduced by employing the effective-boundary-layer theory derived by Bach and Bruus [17]. We assume time-harmonic first-order fields with angular frequency ω=2πf for the acoustic pressure ${\tilde{p}}_{1}\left(r,t\right)=p_{1}\left(r\right)\;e^{-i\omega t}$, the electric potential $\tilde{\varphi }\left(r,t\right)=\varphi \left(r\right)\;e^{-i\omega t}$, and the displacement field $\tilde{u}\left(r,t\right)=u\left(r\right)\;e^{-i\omega t}$. Derived through a perturbation approach, these fields represent tiny perturbations of the unperturbed zero-order fields.

### 2.1. Governing Equations

For a fluid with speed of sound $c_{0}$, density $\rho_{0}$, dynamic and bulk viscosity of the fluid $\eta_{0}$ and $\eta_{0}^{b}$, damping coefficient $\Gamma_{0}$, and the isentropic compressibility $\kappa_{0}={\left(\rho_{0}c_{0}^{2}\right)}^{-1}$, the acoustic pressure $p_{1}$ is governed by the Helmholtz equation, and the acoustic velocity $v_{1}$ is a gradient field,
(1a)$$\begin{array}{cc} \nabla^{2}\;p_{1} & =-\frac{\omega^{2}}{c_{0}^{2}}\left(1+i\Gamma_{0}\right)\;p_{1},\;\mathrm{with}\;\Gamma_{0}=\left(\frac{4}{3}\eta_{0}+\eta_{0}^{b}\right)\;\omega \kappa_{0}, \end{array}$$
(1b)$$\begin{array}{cc} v_{1} & =-i\;\frac{1-i\Gamma_{0}}{\omega \rho_{0}}\;\nabla p_{1}. \end{array}$$

For an elastic solid with density $\rho_{\mathrm{sl}}$, the displacement field u is governed by the Cauchy equation
(2)$$-\omega^{2}\rho_{\mathrm{sl}}\;u=\nabla \cdot \sigma ,$$
where σ is the stress tensor. In the Voigt notation, the 1×6 stress σ and strain s column vectors are given by the 6×1 transposed row vectors $\sigma^{T}=\left(\sigma_{xx},\sigma_{yy},\sigma_{zz},\sigma_{yz},\sigma_{xz},\sigma_{xy}\right)$ and $s^{T}=\left(\partial_{x}u_{x},\partial_{y}u_{y},\partial_{z}u_{z},\partial_{y}u_{z}+\partial_{z}u_{y},\partial_{x}u_{z}+\partial_{z}u_{x},\partial_{x}u_{y}+\partial_{y}u_{x}\right)$, respectively, and σ is related to s by the 6×6 stiffness tensor C having the elastic moduli $C_{ik}$ as components. For a linear, isotropic, elastic solid of the ∞mm-symmetry class the relation is,
(3)$$\sigma =C\cdot s,\;C=\left(\begin{array}{cccccc} C_{11} & C_{12} & C_{13} & 0 & 0 & 0\\ C_{12} & C_{11} & C_{13} & 0 & 0 & 0\\ C_{13} & C_{13} & C_{33} & 0 & 0 & 0\\ 0 & 0 & 0 & C_{44} & 0 & 0\\ 0 & 0 & 0 & 0 & C_{44} & 0\\ 0 & 0 & 0 & 0 & 0 & C_{66} \end{array}\right).$$

Here, the components $C_{ik}=C_{ik}^{\prime }+iC_{ik}^{\prime \prime }$ are complex-valued with real and imaginary parts relating to the speed and the attenuation of sound waves in the solid, respectively. In this work, we assume the glass and the glue layer to be isotropic, yielding the following relations $C_{33}=C_{11}$, $C_{66}=C_{44}$ and $C_{13}=C_{12}=C_{11}-2C_{44}$. This leaves the two independent complex-valued parameters $C_{11}$ and $C_{44}$, relating to the longitudinal and transverse speed of sound and attenuation in the glass and glue layer. For a lead zirconate titanate (PZT) transducer, $C_{66}=\frac{1}{2}\left(C_{11}-C_{12}\right)$, which leaves five independent complex-valued elastic moduli, $C_{11}$, $C_{12}$, $C_{13}$, $C_{33}$, and $C_{44}$.

The electrical potential φ inside the PZT transducer is governed by Gauss’s law for a linear, homogeneous dielectric with a zero density of free charges,
(4)∇·D=∇·(−ε·∇φ)=0,
where D is the electric displacement field and ε the dielectric tensor. Furthermore in PZT, the complete linear electromechanical coupling relating the stress and the electric displacement to the strain and the electric field is given as,
(5a)$$\begin{array}{cc} \left(\begin{array}{c} \sigma \\ D \end{array}\right) & =\left(\begin{array}{cc} C & -e^{T}\\ e & \varepsilon \end{array}\right)\;\left(\begin{array}{c} s\\ E \end{array}\right), \end{array}$$
(5b)$$\begin{array}{cc} \mathrm{with}\;\;e & =\left(\begin{array}{cccccc} 0 & 0 & 0 & 0 & e_{15} & 0\\ 0 & 0 & 0 & e_{15} & 0 & 0\\ e_{31} & e_{31} & e_{33} & 0 & 0 & 0 \end{array}\right)\;\;\mathrm{and}\;\;\varepsilon =\left(\begin{array}{ccc} \varepsilon_{11} & 0 & 0\\ 0 & \varepsilon_{11} & 0\\ 0 & 0 & \varepsilon_{33} \end{array}\right)\;. \end{array}$$

### 2.2. The Acoustic Radiation Force and the Acoustophoretic Particle Velocity

We consider polystyrene particles with density $\rho_{\mathrm{ps}}$, compressibility $\kappa_{\mathrm{ps}}$, and a radius *a*, which is much larger than the viscous boundary layer and much smaller than the acoustic wavelength. In this case, the acoustic radiation force $F^{\mathrm{rad}}$ on the particles placed in water is given by the negative gradient of the Gorkov potential $U^{{\mathrm{rad}}_{}}$, [18]
(6a)$$\begin{array}{cc} F^{\mathrm{rad}} & =-\nabla U^{{\mathrm{rad}}_{}},\;\;\mathrm{with} \end{array}$$
(6b)$$\begin{array}{cc} U^{{\mathrm{rad}}_{}} & =\pi a^{3}\left(\frac{1}{3}f_{0}\;\kappa_{0}{\left|p_{1}\right|}^{2}-\frac{1}{2}f_{1}\;\rho_{0}{\left|v_{1}\right|}^{2}\right),\;\;f_{0}=1-\frac{\kappa_{\mathrm{ps}}}{\kappa_{0}},\;\;\mathrm{and}\;\;f_{1}=\frac{2(\rho_{\mathrm{ps}}-\rho_{0})}{2\rho_{\mathrm{ps}}+\rho_{0}}. \end{array}$$

If a (polystyrene) microparticle of radius *a* is placed in a fluid of viscosity $\eta_{0}$ flowing with the local velocity $v_{0}$, the presence of $F^{\mathrm{rad}}$ imparts a so-called acoustophoretic velocity $v_{\mathrm{ps}}$ to the particle. As inertia is negligible, $v_{\mathrm{ps}}$ is found from a balance between $F^{\mathrm{rad}}$ and the viscous Stokes drag force $F^{{\mathrm{drag}}_{}}$, [14]
(7)$$vAcoustv_{\mathrm{ps}}=\frac{1}{6\pi \eta_{0}a}\;F^{\mathrm{rad}}+v_{0}.$$

### 2.3. Electrical Impedance and Power Dissipation

For a PZT transducer with an excited top electrode and a grounded bottom electrode set by the respective potentials $\varphi =\varphi_{\mathrm{pzt}}$ and φ=0 V, the electrical impedance *Z* is given by the ratio of $\varphi_{\mathrm{pzt}}-0\;$ V and the surface integral of the polarization current density $D+\epsilon_{0}\nabla \varphi$ as [15],
(8)$$Z=\frac{\varphi_{\mathrm{pzt}}}{I},\;\mathrm{with}\;I=-i\omega \int_{\partial \Omega }n\cdot \left(D+\epsilon_{0}\nabla \varphi \right)\;da.$$

The electrical power dissipation $P_{\mathrm{pzt}}$ in the PZT transducer is given by the usual expression
(9)$$P_{\mathrm{pzt}}=\frac{1}{2}Re[\left(\varphi_{\mathrm{pzt}}\right)I^{☆}]=\frac{1}{2}|\varphi_{\mathrm{pzt}}|\;|I|\;\cos\theta ,\;\;\mathrm{with}\;\;\theta =\arg\left(Z\right).$$

### 2.4. Butterworth-Van Dyke Circuit Model

To describe the electrical response of the transducer around its thickness resonance frequency, we use a single-frequency Butterworth-Van Dyke (BVD) model. We furthermore include the impact of the wiring and the parasitic effects of the circuit leading to the PZT transducer in our model. An equivalent circuit of our model is shown in Figure 1a. It consists of the parasitic wire resistance $R_{\mathrm{wire}}$ and inductance $L_{\mathrm{wire}}$ in series with a PZT circuit having the transducer capacitance $C_{0}$ in parallel with an transducer LCR-circuit $R_{1}$-$L_{1}$-$C_{1}$. The four parameters $R_{1}$, $L_{1}$, $C_{1}$, and $C_{0}$ can be obtained from the PZT admittance spectrum Y(f)=1/Z(f) at the resonance frequency $f_{r}$ and anti-resonance frequency $f_{a}$ [19,20],
(10)C0=ImY(fr)2πfr,R1=1ImY(fr),C1=C0fa2fr2−1,L1=1(2πfr)2C1.

We perform simulations of the BVD-circuit using the SPICE-based circuit simulator software LTspice with parameters for the circuit components obtained via Equation (10) and the measured values of the wire resistance $R_{\mathrm{wire}}$ and inductance $L_{\mathrm{wire}}$.

## 3. Materials and Methods

### 3.1. The Experimental Setup

In this work, an acoustofluidic device is used, that consists of a 483-µm-high, 2324-µm-wide, and 50.9-mm-long glass capillary (VitroTubes, VitroCOM, Mountain Lakes, NJ, USA) containing a 200 µm high and 2.06 mm wide microchannel. The device is glued to a cylindrical piezoelectric transducer disk (Pz27, Meggitt A/S, Kvistgaard, Denmark), made from PZT, of thickness 506µm and diameter 10.045 mm with a nominal resonance frequency at around 4 MHz. The capillary tube is glued to the transducer by a thin (39µm) layer of UV-curable glue (NOA 86H, Norland Products, Jamesburg, NJ, USA). An overview of the device is shown in Figure 1b. Using silicone glue, the device is mounted on a 3D-printed sample holder, and rubber tubing is glued to the glass capillary tube on both ends. The electrical connection to the piezoelectric transducer is made via four spring-loaded pins, as can be seen in Figure 1c. These pins both minimize the clamping force on the transducer and enable four-probe measurement of the electric voltage across the transducer.

A schematic overview of the electrical circuit, including signal generator and amplifier, is shown in Figure 1a. As signal generator an Analog Discovery 2 (Digilent, Pullman, WA, USA) in connection with the power amplifier TOE 7607 (TOELLNER Electronic Instrumente GmbH, Herdecke, Germany) is used to drive the piezoelectric transducer. The output of the amplifier is connected to the spring-loaded pins via a coaxial cable followed by 30cm hookup wire. The wire is considered as a short transmission line with negligible capacitance, but with non-negligible parasitic resistance $R_{\mathrm{wire}}$ and inductance $L_{\mathrm{wire}}$. In our simplified circuit model, we only consider the thickness resonance at around 4 MHz of the transducer, and model the transducer via the BVD-model of Equation (10).

### 3.2. Fabrication and Characterization of the Devices

The device is assembled in a step-by-step procedure, and after each fabrication step the electrical impedance spectrum Z(f) of the piezoelectric transducer is recorded with the Vector Network Analyzer Bode 100 (OMICRON electronics GmbH, Klaus, Austria) in the range from 500 Hz to 5 MHz. Device dimensions were measured using an electronic micrometer (RS Pro, RS Components, Corby, UK) with an accuracy of ±4µm. The assembly process consisted of the followings seven steps:1.Measure the dimensions of the capillary tube and the transducer.2.Measure the initial impedance spectrum $Z_{\mathrm{init}}\left(f\right)$ of the Pz27 disk.3.Fit the Pz27 material parameters using ultrasound electrical impedance spectroscopy (UEIS), following the method described in Ref. [21].4.Glue the capillary tube onto the transducer and UV-curing using an exposure time of 168 s at a UV-intensity of $15\;\mathrm{mW}/{\mathrm{cm}}^{2}$ and a wavelength of 365 nm.5.Measure the total device thickness to obtain the glue layer thickness.6.Mount the device on a 3D-printed sample holder and connect rubber tubing using silicone glue7.Measure the impedance spectrum $Z_{\mathrm{sys}}\left(f\right)$ of the combined capillary-glue-transducer system, both air- and fluid-filled.

Using the four-probe setup, shown in Figure 1a,c, the voltage amplitudes $\varphi_{\mathrm{gen}}$ at the signal generator, $\varphi_{\mathrm{amp}}$ at the amplifier, and $\varphi_{\mathrm{pzt}}$ directly across at the transducer were recorded during the measurements. The time-averaged dissipated power $P_{\mathrm{pzt}}$ for a given frequency *f* is the standard expression calculated from Equations (8) and (9) as
(11)$$P_{\mathrm{pzt}}\left(f\right)=\frac{1}{2}\frac{\varphi_{\mathrm{pzt}}^{2}\cos\left[\theta_{\mathrm{sys}}\left(f\right)\right]}{|Z_{\mathrm{sys}}\left(f\right)|},\;\;\mathrm{with}\;\;\theta_{\mathrm{sys}}=\arg\left(Z_{\mathrm{sys}}\right).$$

A feedback control system is implemented to actuate the transducer at the desired constant power or constant voltage during the frequency sweeps. In the following analysis, we consider the following three case:**Case 1:** Constant voltage at the generator, $\varphi_{\mathrm{gen}}=1$ V.**Case 2:** Constant voltage at the transducer, $\varphi_{\mathrm{pzt}}=0.5$ V.**Case 3:** Constant power dissipation in the transducer, $P_{\mathrm{pzt}}=50$ mW.

### 3.3. Determination of Acoustofluidic Resonance Frequencies by Particle Tracking Velocimetry

The acoustofluidic resonance frequencies were determined by measuring the average velocity of particles focusing under acoustofluidic actuation. For the particle focusing experiment, we used a neutrally buoyant solution of 10-μm-diameter fluorescent polystyrene spheres (microparticles GmbH, Berlin, Germany) in a fluid consisting of 83% (*v/v*) ultrapure water (Direct-Q3 System, Merck) and 17% (*v/v*) OptiPrep (Density Gradient Medium, Sigma-Aldrich). The particle concentration was approximately 500 particles/µ L.

The acoustic focusing was studied using single-camera 3D particle tracking performed with the general defocusing particle tracking (GDPT) method [22]. GDPT determines the depth position of defocused particle images from the analysis of the corresponding defocusing patterns, previously mapped with a proper calibration procedure [23]. The particle images were recorded using a high-sensitive sCMOS camera (pco.edge 5.5, Excelitas PCO GmbH, Kelheim, Germany) with an optical system consisting of a 5× microscope objective (EC EPIPlan, Zeiss AG, Oberkochen, Germany) and a cylindrical lens in front of the camera sensor to enhance the defocusing patterns. The images were processed using the open-source software *defocustracker* version 2.0.0 [24].

In each experiment, 200 images were recorded at 25 frames/s, and the signal generator was switched on precisely 1 s after the camera had started to record the first frame using an electrical trigger. The frequency sweeps were performed at frequencies in the range from 3.3 to 4.3 MHz in steps of 10 kHz. After the GDPT evaluation, we obtained a set of *N* measured three-dimensional particle trajectories $s^{\left(j\right)}\left(t\right)={\left\{x\left(t\right),y\left(t\right),z\left(t\right)\right\}}^{\left(j\right)}$ for each frequency. We then proceeded to compute the three components $s_{i}^{\exp}\left(t\right)$ of the average cumulative particle displacement vector $s^{\exp}\left(t\right)$ as
(12)$$s_{i}^{\exp}\left(t\right)=\frac{1}{N}\sum_{j=1}^{N}\left|s_{i}^{\left(j\right)}\left(t\right)-s_{i}^{\left(j\right)}\left(t_{0}\right)\right|,\;\mathrm{for}\;i=x,y,z,$$
where $t_{0}$ is the time when the acoustics was turned on. The average acoustophoretic speed $v_{\exp}$ of the particles was then calculated at time $t_{\exp}=40\;\mathrm{ms}$ after turning on the acoustics,
(13)$$v_{\exp}=\sqrt{\sum_{i=x,y,z}{\left(\partial_{t}s_{i}^{\exp}{|}_{t=t_{\exp}}\right)}^{2}}.$$

## 4. Numerical Model

### 4.1. Description of the Modeled System

We perform numerical simulations of the device described in Section 3.1 using the software COMSOL Multiphysics 6.0, following the implementation in Refs. [3,14,25]: We use the weak form interface “*Weak Form PDE*” to obtain the potential φ in the PZT, the displacement u in all solids, and the acoustic pressure field $p_{1}$ in the fluid channel. By using the *x*-*z* and *y*-*z* symmetry planes, only a quarter of the actual geometry is modeled. In the model we consider the piezoelectric transducer, a thin coupling layer, and the water-filled glass capillary tube. To further minimize the computational complexity, we apply a perfectly matched layer (PML) at the end of the glass capillary [25]. The PML mimicks perfect absorption of all outgoing waves, and it allows to reduce the length of the capillary tube. In our experimental setup, the damping at the edge of the tube is ensured by the silicone glue connecting the tube to the sample holder. A sketch of the system is shown in Figure 1d,e. The simulations were performed on a workstation with a 12-core, 3.5-GHz central processing unit and 128 GB random access memory. Details on the mesh convergence analysis and the material parameters used for the simulation can be found in the Appendix A and Appendix B.

### 4.2. Numerical Simulation of the Particle Velocity

In the second and final study step of our simulation model, we use the “*Particle Tracing for Fluid Flow*” module [26] to compute the particle trajectories of 1000 randomly distributed particles based on the fields computed in the first study step. The wall condition is set to “*Stick*” to mimic stuck particles, which were also commonly observed in the experimental setup. The force acting on the particles is the simulated radiation force $F^{\mathrm{rad}}$, see Equation (6). Acoustic streaming is neglected as the particles are relatively big [26], and the influence of gravity is neglected due to the use of a neutrally-buoyant solution. Similarly to what is done for the experimental data, we obtain the velocity $v_{i}$ of each particle *i* at time $t_{\mathrm{sim}}$ and compute the average speed $v_{\mathrm{sim}}$ of the particle as
(14)$$v_{\mathrm{sim}}=\frac{1}{N}\sum_{i=1}^{N}|v_{i}\left(t_{\mathrm{sim}}\right)|,$$
at time $t_{\mathrm{sim}}=\left(40\;t_{\mathrm{sim}}^{\mathrm{foc}}/t_{\exp}^{\mathrm{foc}}\right)\;\mathrm{ms}$, where $t_{\mathrm{sim}}^{\mathrm{foc}}/t_{\exp}^{\mathrm{foc}}$ is the ratio between the numerical and experimental focusing time at resonance.

### 4.3. Boundary Conditions at the Fluid-Solid and the PZT-Solid Interfaces

In the simulations, we assume a time-harmonic voltage amplitude of $\varphi_{\mathrm{pzt}}$ at the top surface of the piezoelectric transducer, while the bottom surface is grounded to $\varphi_{\mathrm{gnd}}=0$. We furthermore assume continuous stress between the different domains, a zero normal component of the dielectric displacement field D·n=0 at the PZT-air interfaces and zero normal stress at the solid-air interfaces [14]. For the fluid-solid interface, we implement the effective boundary conditions derived by Bach and Bruus [17]. Here, the fields inside the very thin boundary layers of thickness $\delta_{\mathrm{fl}}=\sqrt{2\eta_{0}/\left(\rho_{0}\omega \right)}\approx 0.5\;µm$ are taken into account analytically. The pressure $p_{1}$ at the fluid-air interface is set to zero. The boundary conditions between the different domains and their corresponding boundary are summarized in Table 1.

We assume symmetry of all simulated fields at the yz-plane at x=0 and at the xz-plane at y=0. The symmetry boundary conditions therefore are implemented as follows:
(15a)$$\begin{array}{cccccccc} \mathrm{Symmetry}\;\mathrm{at}\; & x=0: & & \\ u_{x} & =0,\; & \sigma_{yx} & =\sigma_{zx}=0,\; & \partial_{x}p_{1} & =0,\; & \partial_{x}\varphi & =0. \end{array}$$
(15b)$$\begin{array}{cccccccc} \mathrm{Symmetry}\;\mathrm{at}\; & y=0: & & \\ u_{y} & =0,\; & \sigma_{xy} & =\sigma_{zy}=0,\; & \partial_{y}p_{1} & =0,\; & \partial_{y}\varphi & =0. \end{array}$$

## 5. Results and Discussion

### 5.1. Electrical Impedance Measurements

The electrical impedance spectrum Z(f) of the unloaded and loaded Pz27 transducer was measured after each step in the fabrication procedure. The material parameters of the specific Pz27 transducer used in this study, were obtained by electrical impedance spectroscopy (UEIS), following the procedure described in Ref. [21], based on the measured Z(f) of the unloaded transducer. The result is shown in Figure 2a, were it is seen that the fitted spectrum |Z(f)| agrees well with the measured one. The piezoelectric parameters obtained from this UEIS fitting, were then subsequently used together with the remaining material parameters listed in Appendix B to simulate numerically the pressure field $p_{1}$, the displacement field u, and the electric potential φ of the transducer-glue-capillary-tube system. The measured and the simulated impedance spectrum of the full system in the frequency range 3.3 to 4.3 MHz are shown in Figure 2b, and good agreement is found. Numerical simulations were performed using both the fitted values, and the values for Pz27 given in Ref. [21]. The discrepancy between the two resulting spectra emphasizes the need of obtaining fitted material parameters for the specific transducer used in the study. The remaining deviations from the measured impedance spectrum stem from uncertainties in the glass material parameters, which where taken from the literature and not fitted by UEIS.

### 5.2. Impact of the Cable and the Circuit Resonances on Measured Voltage Amplitude

Using the measured impedance spectrum Z(f) of the full system around the 4-MHz resonance, we obtain the required parameters to describe the transducer through the BVD model of Section 2.4, and we find $C_{0}=1.28$ nF, $C_{1}=207\;$ pF, $L_{1}=7.69\;$µH, and $R_{1}=2.59\;\Omega$. We estimate for each of the two wires connecting the amplifier and the transducer that $R_{\mathrm{wire}}=1\;\Omega$ and $L_{\mathrm{wire}}=411\;$ nH. In Figure 2c, we compare Z(f) computed from the BVD model with the measured Z(f). It is seen that the BVD model captures well the characteristics around the 4-MHz resonance of the transducer, and it can therefore aid the understanding of the circuit characteristics.

When comparing the voltage amplitudes at various points in the circuit using a constant generator voltage amplitude $\varphi_{\mathrm{gen}}=1\;$ V, we find as shown in Figure 2d that for most of the frequencies in the range 0.5–5000 kHz, the voltage amplitude $\varphi_{\mathrm{pzt}}$ across the transducer is larger than the voltage amplitude $\varphi_{\mathrm{amp}}$ right after the amplifier. This may seem counter-intuitive, but given the resonant nature of the circuit, charge may build up on the capacitive circuit elements. We furthermore find two frequencies were the voltage amplitude is minimal: At $f_{\mathrm{amp}}=3.56\;$ MHz, the impedance of the full circuit has an impedance minimum, and at $f_{\mathrm{pzt}}=3.98\;$ MHz, the impedance of the transducer has a minimum. The down-shift of $f_{\mathrm{amp}}$ by 0.42 MHz from $f_{\mathrm{pzt}}$ is due to the parasitic inductance of the wire connecting the amplifier and the transducer. We note that if the voltage amplitude is recorded right after the amplifier, and not directly across the transducer, a wrong estimate of the voltage amplitude and power dissipation of the transducer may result. Furthermore, the parasitic inductance minimizes the power transfer from the amplifier to the transducer, and therefore it is in general beneficial to minimize this inductance by use of shortened and shielded cables. To minimize the ratio $r=f_{\mathrm{amp}}/f_{\mathrm{pzt}}$, the inductance of the wire $L_{\mathrm{wire}}$ should be minimized according to
(16)$$L_{\mathrm{wire}}<\frac{\left(1-r^{2}\right)}{2r^{2}}\frac{C_{1}L_{1}}{C_{1}+C_{0}\left(1-r^{2}\right)}.$$

In our circuit, it is required that $L_{\mathrm{wire}}<70\;$ nH to keep the mismatch of $f_{\mathrm{circuit}}$ and $f_{\mathrm{pzt}}$ below 1%. This is typically hard to achieve, as it requires very thin and short wires. Alternatively, a capacitor $C_{\mathrm{comp}}={\left(2\pi f_{\mathrm{comp}}\right)}^{-2}L_{\mathrm{wire}}^{-1}$ in series with the wire could be used to counteract the impact of $L_{\mathrm{wire}}$ at the frequency $f_{\mathrm{comp}}$. Further improvements of the circuit could be obtained by impedance-matching the load impedance $Z_{\mathrm{load}}$ to the source impedance $Z_{\mathrm{source}}$, by adding circuit components to the load such that $Z_{\mathrm{source}}=Z_{\mathrm{load}}^{*}$ [19].

### 5.3. Voltage Amplitude and Power Dissipation

The voltage amplitudes and power dissipation in a frequency sweep are depending significantly on the chosen electrical excitation method in the actuation process. In Figure 3a–c are shown frequency sweeps of the the voltage amplitudes $\varphi_{\mathrm{gen}}$ and $\varphi_{\mathrm{pzt}}$, and the average dissipated power $P_{\mathrm{pzt}}$ for the three considered cases of constant $\varphi_{\mathrm{gen}}$, constant $\varphi_{\mathrm{pzt}}$, and constant $P_{\mathrm{pzt}}$.

The first case with constant $\varphi_{\mathrm{gen}}=1\;$ V is shown in Figure 3a. It is seen that $\varphi_{\mathrm{pzt}}$ and $P_{\mathrm{pzt}}$ are minimal at or close to the transducer impedance minimum at $f_{\mathrm{pzt}}=3.98$ MHz. This is not an ideal situation, because many systems are designed with a resonance close to the nominal transducer resonance in mind. Instead, it is beneficial to stabilize either $\varphi_{\mathrm{pzt}}$ or $P_{\mathrm{pzt}}$.

The second case with constant $\varphi_{\mathrm{pzt}}=0.5$ V is shown in Figure 3b. Both $\varphi_{\mathrm{gen}}$ and $P_{\mathrm{pzt}}$ have a narrow peak near $f_{\mathrm{pzt}}=3.98$ MHz. As will be discussed in Section 5.4 and Section 5.5, this might not be ideal for particle focusing experiments when comparing device performances over wider frequency ranges.

The third case with constant $P_{\mathrm{pzt}}=50\;$ mW is shown in Figure 3c. To stabilize $P_{\mathrm{pzt}}$, the voltage $\varphi_{\mathrm{gen}}$ needs to be adjusted according to the electrical impedance spectrum of the transducer. The voltage $\varphi_{\mathrm{gen}}$ needs to be higher when running the transducer on-resonance, compared to off-resonance. As in the first case, the voltage amplitude $\varphi_{\mathrm{pzt}}$ is minimal at the resonance $f_{\mathrm{pzt}}=3.98$ MHz. Other effects, such as the non-linear gain of the amplifier and non-linearity of the piezoelectric transducer, may lead to increased discrepancies between $\varphi_{\mathrm{gen}}$ and $\varphi_{\mathrm{pzt}}$, which furthermore emphasizes the need to monitor $\varphi_{\mathrm{pzt}}$ and $P_{\mathrm{pzt}}$, and to specify which of them, if any, is kept constant.

### 5.4. Average Acoustophoretic Particle Speed

When the acoustic pressure field $p_{1}$ is switched on via the Pz27 transducer in our setup shown in Figure 1, polystyrene microparticles inside the water-filled capillary tube acquire an acoustophoretic velocity $v_{\mathrm{ps}}$, see Equations (7), (13) and (14), proportional to the acoustic radiation force $F^{\mathrm{rad}}$, see Equation (6). For the above three cases, the experimental results for the average particle speed $v_{\exp}$ are shown in Figure 4a, and the corresponding results for $P_{\mathrm{pzt}}$ are shown in Figure 4b. In the case of constant $\varphi_{\mathrm{gen}}=1\;$ V, we measure the highest particle speed at the resonance frequency $f=f_{\exp}^{\mathrm{res},g}=4.24\;$ MHz with $v_{\exp}\approx 46\;µm\;s^{-1}$. In the following discussion, we refer to experimental $f_{\exp}^{(ny,nz)B}$ and simulated $f_{\mathrm{sim}}^{(ny,nz)B}$ resonance frequencies with well-identified numbers $n_{y}$ and $n_{z}$ of standing half-waves in the width (*y*) and height (*z*) directions of the microchannel in the respective cases of constant $\varphi_{\mathrm{gen}}$, $\varphi_{\mathrm{pzt}}$, and $P_{\mathrm{pzt}}$ indicated by the superscript “B” = “g”, “φ”, and “*P*”. For constant $\varphi_{\mathrm{pzt}}=0.5\;$ V, we find the highest particle speed at $f=f_{\exp}^{(0,1)\varphi }=3.98\;$ MHz with $v_{\exp}\approx 71\;µm\;s^{-1}$. Finally, for constant $P_{\mathrm{pzt}}=50\;$ mW, a resonance appears at $f=f_{\exp}^{(0,1)P}=4.03$ MHz with $v_{\exp}\approx 77\;µm\;s^{-1}$. Furthermore, for $\varphi_{\mathrm{pzt}}=0.5\;$ V and $P_{\mathrm{pzt}}=50\;$ mW, we observe a local resonance at $f=f_{\exp}^{(10,0)P}=3.51\;$ MHz. This frequency $f_{\exp}^{(10,0)P}$, corresponding to $n_{y}=10$ half-wavelengths across the channel width, relates to an acoustic mode leading to particle focusing in 10 nodal lines parallel to the length and evenly distributed across the width of the microchannel, as discussed further in Section 5.5.

When analyzing the measurements in Figure 4b of the power dissipation in the three cases, we find that, when driving the transducer at constant $\varphi_{\mathrm{pzt}}$, a clear maximum in $P_{\mathrm{pzt}}$ appears at $f_{\exp}^{(0,1)\varphi }=3.98\;$ MHz, but conversely, $P_{\mathrm{pzt}}$ has a minimum at the same frequency for constant $\varphi_{\mathrm{gen}}$. The reason is that at this frequency, the transducer has an intrinsic resonance and thus a minimum in its impedance. Lastly, we note that experimentally it is difficult to perfectly stabilize $P_{\mathrm{pzt}}$ near the transducer resonance $f_{\exp}^{(0,1)\varphi }=3.98\;$ MHz. This difficulty is likely due to on-resonance heating effects of the transducer.

### 5.5. Comparing Numerical Simulations with Experiments

In Figure 4c using constant $P_{\mathrm{pzt}}=50\;\mathrm{mW}$, the measured $v_{\exp}^{P}/v_{\max}^{P}$ and the simulated $v_{\mathrm{sim}}^{P}/v_{\max}^{P}$ average acoustophoretic speed, normalized by the measured maximum speed $v_{\max}^{P}=\max\left|v_{\exp}^{P}\right|$, are plotted versus frequency for 3.3–4.3 MHz. The agreement between the two curves is good, and they both show a resonance at nearly the same frequency $f=f_{\exp}^{(0,1)P}=4.03\;$ MHz and $f=f_{\mathrm{sim}}^{(0,1)P}=4.04\;$ MHz, respectively. A similar plot is shown in Figure 4d, but now for the case of constant $\varphi_{\mathrm{pzt}}=0.5\;V$, namely the measured $v_{\exp}^{\varphi }/v_{\max}^{P}$ and the simulated $v_{\mathrm{sim}}^{\varphi }/v_{\max}^{P}$ versus frequency with the same normalization $v_{\max}^{P}$ as before. Again, the agreement between simulation and experiment is good, and both curves have a maximum at $f=f_{\exp}^{(0,1)\varphi }=f_{\mathrm{sim}}^{(0,1)\varphi }=3.98\;$ MHz, about 50 kHz lower than the constant-power resonance frequency $f_{\exp}^{(0,1)P}=4.03\;$ MHz.

Some interesting features are seen in the measured and simulated spectrum of the constant-voltage acoustophoretic velocity spectrum $v_{\exp}^{\varphi }\left(f\right)$ in Figure 4d. Its maximum, obtained at $f_{\exp}^{(0,1)\varphi }=3.98\;$ MHz, is 8% less then the one obtained in the constant-power velocity spectrum $v^{P}\left(f\right)$ in Figure 4c, $v_{\max}^{\varphi }=0.92\;v_{\max}^{P}$. Moreover, far from being at the maximum, $v^{\varphi }\left(f_{\exp}^{P}\right)=0.27\;v_{\max}^{P}$ is a local minimum. Clearly, the optimal operating condition for acoustophoresis is to run the system at $f_{\exp}^{(0,1)P}$ with constant-power actuation. Operating directly at the transducer resonance at $f_{\exp}^{(0,1)\varphi }=3.98\;$ MHz, is not equally efficient due to the low impedance of the transducer and the resulting high power dissipation at this frequency. Constant-voltage frequency sweeps can be misleading in that regard.

In the simulation and experiment with constant $P_{\mathrm{pzt}}$, see Figure 4a,c, a local maximum in the average particle speed *v* is observed at $f=f_{\exp}^{(10,0)P}=3.51\;$ MHz. In Figure 5a,b we compare the simulated pressure field at this frequency with the simulation results at $f=f_{\exp}^{(0,1)P}=4.03\;$ MHz. Images of the particles after 4 s at the two corresponding frequencies are shown in Figure 5c,d. Both in the numerically simulated pressure field, as well as in the measured particle positions, we observe in the *x*-*y* plane at $f=f_{\exp}^{(10,0)P}=3.51\;$ MHz, the formation of 10 nodal lines parallel to the tube axis along the *x* direction, and with an equidistant distribution across the width, see Figure 5c. In contrast, at the main resonance at $f_{\exp}^{(0,1)P}=4.03\;$ MHz shown in Figure 5d, we observe particle focusing in the *x*-*y* center plane of the glass capillary tube, above the center region of the Pz27 transducer, caused by the standing half-wave in the vertical *z*-direction. No transverse nodal lines are observed here.

## 6. Conclusions

Monitoring power dissipation in and voltage across the piezoelectric transducer is important and helpful for understanding and optimizing the performance of acoustofluidic systems. As shown by our measurements on the setup shown in Figure 1, the voltage can differ significantly between amplifier output and transducer, due to the varying impedance of the transducer at different frequencies. In this work, we compared the performance of an acoustofluidic device using three types of actuation: (Case 1) Supplying a constant voltage amplitude $\varphi_{\mathrm{gen}}$ to the amplifier input from the signal generator, (Case 2) driving the piezoelectric transducer using constant-voltage actuation $\varphi_{\mathrm{pzt}}$, and (Case 3) keeping the power dissipation $P_{\mathrm{pzt}}$ in the transducer constant. The acoustofluidic performance was evaluated in terms of the average acoustophoretic particle speed *v* in the microfluidic channel measured with 3D particle tracking velocimetry and computed numerically, see Figure 4.

Case 1, performing frequency sweeps with constant $\varphi_{\mathrm{gen}}$, which is typically used for acoustofluidic devices, may result in a misleading identification of the ideal actuation frequency. The reason is that the power dissipation in the transducer is dependent of the impedance of the transducer as well as the resonant behavior of the cables connecting amplifier and transducer. Instead, keeping a constant power $P_{\mathrm{pzt}}$ is a better choice for obtaining a reproducible characterization of the intrinsic properties of acoustofluidic devices.

Case 2, frequency sweeps with constant $\varphi_{\mathrm{pzt}}$ often result in high power dissipation at the transducer resonance frequency $f_{\exp}^{(0,1)\varphi }$, where the impedance of the transducer is at a minimum. Therefore, the strongest acoustofluidic response will be observed closest to this frequency, but it is likely not the most power-efficient frequency, as it results in increased heating and comes at the cost of high input powers. Acoustofluidic applications, however, are often constrained by power-limitations of the frequency generator or the amplifier, as well as the requirement of maintaining a defined temperature to enable the processing of biological samples.

Case 3, frequency sweeps with constant $P_{\mathrm{pzt}}$ appear to be a better measure to compare device performance across frequencies, as this compensates for the decrease in impedance at the transducer resonance. As a consequence, also finer details in the acoustic fields that occur at frequencies further away from the transducer resonance frequency can be observed. This is exemplified by the transverse resonance in the width direction at $f=f_{\mathrm{sim}}^{(10,0)P}=3.51\;$ MHz, see Figure 5a,c; a resonance clearly visible as a strong peak in the constant-power spectrum in Figure 4c, but not visible in the constant-voltage spectrum in Figure 4d. Keeping $P_{\mathrm{pzt}}$ constant, enhances the intrinsic properties of the device performance, as it does not depend on the wiring. In conclusion, frequency sweeps with constant power are the recommended procedure for the characterization of acoustofluidic resonances across the frequency spectrum, both experimentally and numerically. If it is not possible to control the power, and if the frequency sweeps are performed with constant input voltage at the generator or at the transducer, special care must be taken in analyzing the results to avoid misinterpretation of the data.

Lastly, the external circuitry, see Figure 1, may have an impact on the resonance behavior of the setup. In this work, the parasitic impact of the wire inductance, connecting the amplifier and the transducer, was observed. We note that by fine-tuning the impedance of the external circuitry to match the impedance of the transducer at resonance, the power transfer to the transducer can be increased. Such an impedance matching is common in many other fields. Considering the whole circuit, rather than just the piezoelectric transducer in an acoustofluidic setup, therefore can be beneficial to further improve system performance in various acoustofluidic applications.

## Figures and Tables

**Figure 1 micromachines-13-01886-f001:**
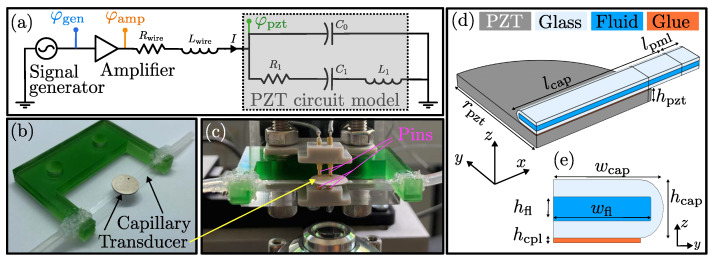
(**a**) A schematic overview of the electrical circuit driving the transducer. The transducer, represented by the BVD-model with a resistor $R_{1}$, an inductor $L_{1}$, and two capacitors $C_{0}$ and $C_{1}$, is coupled in series with the parasitic wire resistance and inductance. (**b**) A disk-shaped piezoelectric transducer is glued to a long, straight glass capillary tube. The tube is connected to a 3D-printed sample holder (green), and inlet/outlet tubing is glued to the ends of the tube. (**c**) The acoustofluidic device is mounted above the microscope lens and is electrically connected via two spring-loaded pins on each side of the transducer. (**d**) Using the symmetry planes *x*-*z* and *y*-*z*, only a quarter of the actual geometry needs to be simulated numerically. The different domains of the model are: PZT (gray), glass (light blue), water (dark-blue), and the thin glue layer (orange). The dimensions are $r_{\mathrm{pzt}}=5.02$ mm, $h_{\mathrm{pzt}}=506\;µm$, $w_{\mathrm{fl}}=2060\;µm$, $h_{\mathrm{fl}}=200\;µm$, $h_{\mathrm{cpl}}=39\;µm$, $w_{\mathrm{cap}}=2324\;µm$, and $h_{\mathrm{cap}}=483\;µm$. In the simulation the reduced lengths are $l_{\mathrm{cap}}=6.44$ mm and $l_{\mathrm{pml}}=839\;µm$. (**e**) The cross-section in the *y*-*z*-plane is showing the glass tube, the water, and the glue layer.

**Figure 2 micromachines-13-01886-f002:**
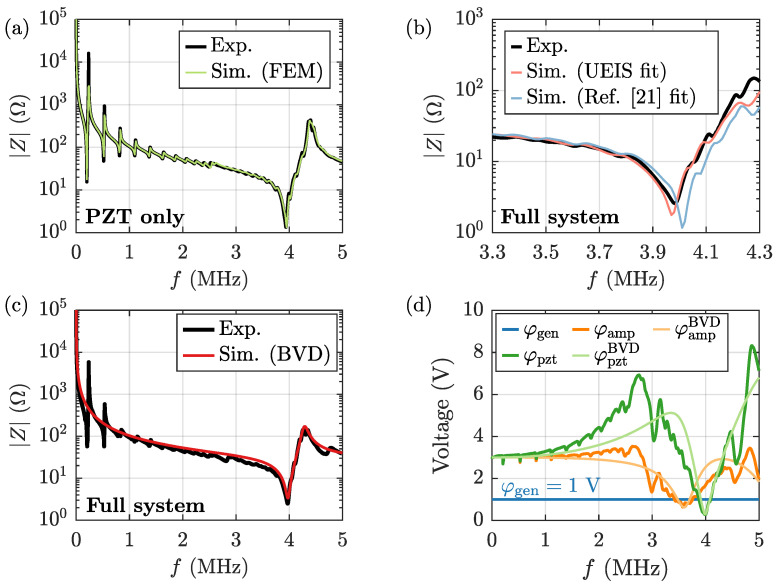
(**a**) Measured (black) and simulated (green) electrical impedance spectrum |Z(f)| in the frequency range 0.5–5000 kHz of the unloaded Pz27 disk. (**b**) Measured (black) and simulated |Z(f)| for 3.3–4.3 MHz, using either UEIS fitted parameters (red) or the parameters from Ref. [21] (blue) of the full system consisting of Pz27 disk, glue layer and liquid-filled glass capillary tube. (**c**) The measured |Z(f)| (black) for 0.5–5000 kHz of the full system (Pz27 disk, glue layer, and liquid-filled glass capillary tube), and the computed |Z(f)| (red) based on the single-frequency BVD-model of Section 2.4. (**d**) The three measured voltage amplitudes versus frequency *f* at different points in the circuit: $\varphi_{\mathrm{pzt}}$ (dark-green) obtained by four-probe measurements directly across the piezoelectric transducer, $\varphi_{\mathrm{amp}}$ (dark-orange) measured at the output of the amplifier, $\varphi_{\mathrm{gen}}$ (dark-blue) measured at the signal generator. Shown also are the two *LTspice*-simulated voltage amplitudes: $\varphi_{\mathrm{pzt}}^{\mathrm{BVD}}$ (light green) and $\varphi_{\mathrm{amp}}^{\mathrm{BVD}}$ (light orange) computed from the BVD-model.

**Figure 3 micromachines-13-01886-f003:**
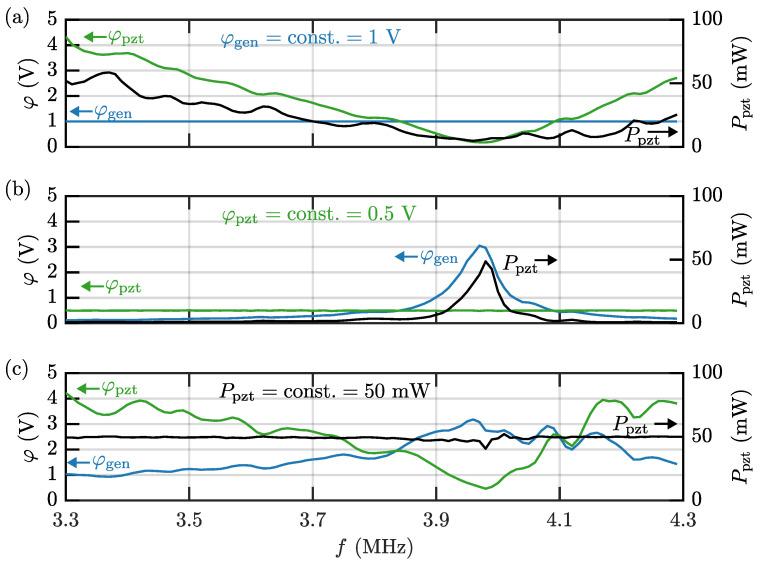
The measured voltage amplitudes $\varphi_{\mathrm{pzt}}$ (green, left axis) and $\varphi_{\mathrm{gen}}$ (blue, left axis) as well as the measured average power dissipation $P_{\mathrm{pzt}}$ (black, right axis) plotted versus frequency for 3.3–4.3 MHz for the three cases (**a**) constant voltage $\varphi_{\mathrm{gen}}$ at the signal generator, (**b**) constant voltage $\varphi_{\mathrm{pzt}}$ across the piezoelectric transducer, and (**c**) $P_{\mathrm{pzt}}$ constant average power dissipation.

**Figure 4 micromachines-13-01886-f004:**
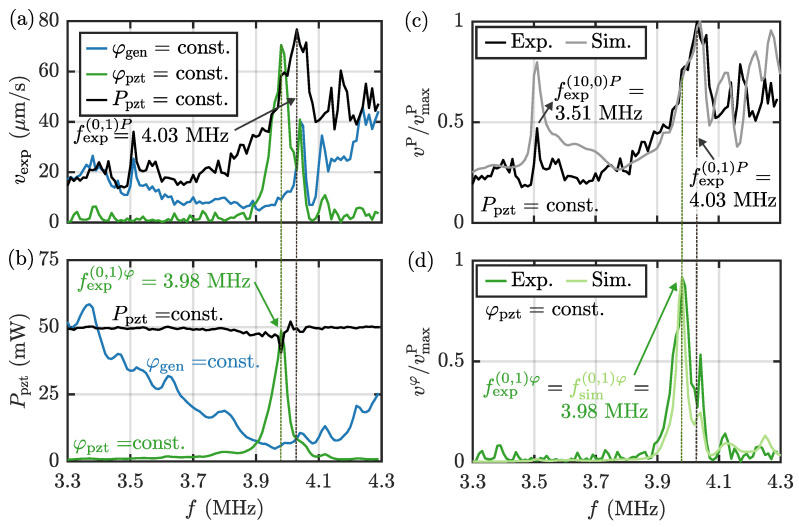
(**a**) Plot of the measured average acoustophoretic particle speed $v_{\exp}$ versus frequency *f* in the range 3.3–4.3 MHz. (**b**) The measured power dissipation $P_{\mathrm{pzt}}$ in the transducer versus *f* for 3.3–4.3 MHz for the three cases of constant voltage $\varphi_{\mathrm{gen}}$ at the signal generator, constant voltage $\varphi_{\mathrm{pzt}}$ across the piezoelectric transducer, and constant average power dissipation $P_{\mathrm{pzt}}$. (**c**) The experimental (black) and simulated (gray) normalized average particle speed $v^{P}/v_{\max}^{P}$ versus frequency *f* at constant-power dissipation $P_{\mathrm{pzt}}=50\;$ mW. (**d**) The experimental (dark-green) and simulated (light-green) normalized average particle speed $v^{\varphi }/v_{\max}^{P}$ versus frequency *f* at constant transducer voltage $\varphi_{\mathrm{pzt}}=0.5\;$ V.

**Figure 5 micromachines-13-01886-f005:**
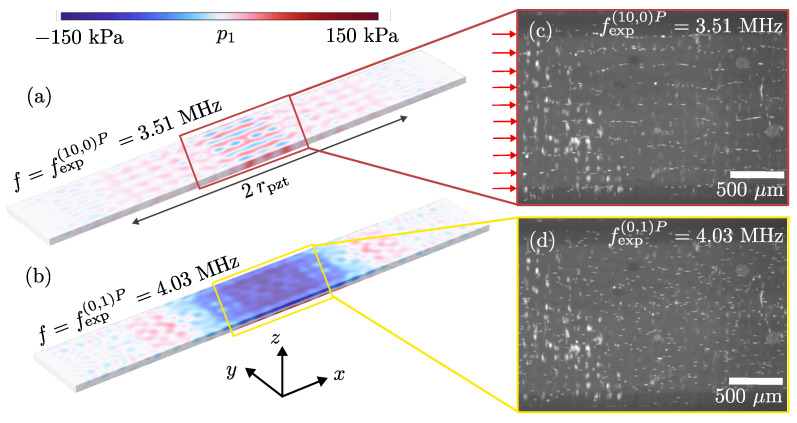
Color plot from −150 kPa (blue) to 150 kPa (red) of the simulated acoustic pressure field $p_{1}$ inside the fluid channel at (**a**) $f=f_{\exp}^{(10,0)P}=3.51\;$ MHz with a standing-5-wavelength resonance mode in the *y*-direction (10 nodal lines above the center region of the transducer), and (**b**) at $f=f_{\exp}^{(0,1)P}=4.03\;$ MHz with a standing-$\frac{1}{2}$-wavelength resonance mode in *z*-direction. (**c**) Micrograph of the particles focused in 10 nodal lines (marked by red arrows) inside the microfluidic channel after 4 s at the resonance frequency $f_{\exp}^{(10,0)P}=3.51\;$ MHz. (**d**) Micrograph of the particles focused in 1 nodal plane (the xy-plane) inside the microfluidic channel after 4 s at the resonance frequency $f_{\exp}^{(0,1)P}=4.03\;$ MHz.

**Table 1 micromachines-13-01886-t001:** The boundary conditions used in the numerical simulations with the surface normal vector n pointing away from the respective domain. We use the solid velocity $v_{\mathrm{sl}}=-i\omega u$, and the complex-valued shear-wave number $k_{s}=\left(1+i\right)\;\delta_{\mathrm{fl}}^{-1}=\left(1+i\right)\;\sqrt{\rho_{0}\omega /\left(2\eta_{0}\right)}$.

Domain	←	Boundary	Boundary Condition
PZT	←	top electrode	$\varphi =\varphi_{\mathrm{pzt}}$
PZT	←	bottom electrode	φ=0
PZT	←	air	D·n=0
Solid	←	air	σ·n=0
Solid	←	fluid	$\sigma \cdot n=-p_{1}\;n+ik_{s}\eta_{0}(v_{\mathrm{sl}}-v_{1})$
Fluid	←	solid	$v_{1}\cdot n=v_{\mathrm{sl}}\cdot n+\frac{i}{k_{s}}\nabla_{\|}\cdot {\left(v_{\mathrm{sl}}-v_{1}\right)}_{\|}$
Fluid	←	air	$p_{1}=0$

## Data Availability

The data presented in this study are available on request from the corresponding author.

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
