# Peer review of "Constant-Power versus Constant-Voltage Actuation in Frequency Sweeps for Acoustofluidic Applications"

_micromachines, 2022, doi:10.3390/mi13111886_

Round 1

Reviewer 1 Report

1I think the title needs to be reformulated to become more "friendly.

2.  Although the language is quite good, the author is advised to check the writing carefully to eliminate some inaccurate sentences and expressions.

3.   In page 5, authors must argue how the equation (11) was obtained, or indicate a bibliographic reference for their original form.

4. In page 7 (Section: 4.2. Numerical simulation of the particle velocity), the authors in simulation model used the "Particle Tracing for Fluid Flow" module and compute the particle trajectories of 1000 randomly distributed particles. However, i think it needs more clarification to know the difference between the introduced module and that modules in the previous studies.

Reviewer 2 Report

The manuscript by et al. showed that a comparison of the effects of different electrical excitation methods on a transducer in a given acoustofluidic device. They investigated the impact of three different actuation approaches during a frequency sweep: (1) Constant input voltage into the amplifier, (2) constant voltage at the transducer, and (3) constant power dissipation in the transducer, and compared experimental findings with their numerical model.

This work is written in a logical sequence, and the conclusions are appropriately supported by the data. I therefore recommend accepting this paper for publication in Micromachines after the authors addressed the following points:

(1)   In page 10, what is the f10width mean? Could the author give a detailed explanation?

(2)   The directivity of the arrows in Figure 3 is not very clear. Could the author improve it?

(3)   The guidelines in Section 6 are not very clear. Could the author suggest usage different scenarios where the three methods are applicable?

(4)   In figure caption of Figure 1, please delete repeating “The dimensions”.
